# Wearable and Washable MnO_2_−Zn Battery Packaged by Vacuum Sealing

**DOI:** 10.3390/nano13020265

**Published:** 2023-01-07

**Authors:** Jun Ho Noh, Myoungeun Oh, Sunjin Kang, Hyeong Seok Lee, Yeong Jun Hong, Chaeyeon Park, Raeyun Lee, Changsoon Choi

**Affiliations:** 1Department of Energy and Materials Engineering, Dongguk University, 30 Pildong-ro, 1-gil, Jung-gu, Seoul 04620, Republic of Korea; 2Department of Advanced Battery Convergence Engineering, Dongguk University, 30 Pildong-ro, 1-gil, Jung-gu, Seoul 04620, Republic of Korea; 3Research Center, Sillo Incorporation, 30 Pildong-ro, 1-gil, Jung-gu, Seoul 04620, Republic of Korea

**Keywords:** textile battery, battery packaging, vacuum sealing, carbon nanotube, MnO_2_−Zn battery

## Abstract

Batteries are used in all types of electronic devices from conventional to advanced devices. Currently, batteries are evolving in the direction of extremely personalized yarn− or textile−structured textronic systems. However, the absence of a protective layer on such batteries is a critical limitation to their practical use. In this study, we developed a wearable and washable MnO_2_−Zn textile battery that maintains its electrochemical capacity under various external environmental conditions through a vacuum−sealed packaging. The packaged textile battery was fabricated by vacuuming a polymer envelope containing the battery, followed by heat sealing with a vacuum packaging machine. The interior and exterior regions of the textile battery are completely separated by the packaging sheath to preclude leakage and intrusion of substances. The resulting packaged textile battery exhibits stable capacity retention performance under varying temperature and humidity; mechanical deformations due to bending, twisting, rubbing, and pressing; and several mechanical, chemical, and their combined washing cycles. On the basis of these demonstrations, we expect that our vacuum−packaged textile battery will offer new possibilities for practical and convenient use of textronics.

## 1. Introduction

Batteries are used in all types of electronic devices, from conventional devices such as sensors, cameras, displays, and laptops to advanced devices such as smartphones, electric vehicles, and wearable devices [1,2,3,4,5]. Energy supplement systems have shifted from centralized to portable supplies with advancements in electronic engineering and are currently evolving into extremely personalized textronic battery systems that have the appearance of advanced wearable devices [6,7,8,9,10]. Accordingly, yarn− or textile−structured batteries with high flexibility, light weight, and wearability have been actively studied in recent years.

Conventional battery systems are rigid, heavy, and bulky, so innovative structures and advanced materials must be introduced to realize textronic batteries, which are a new paradigm in energy supply systems [9,11,12,13,14,15]. One−dimensional yarn−structured batteries have attracted attention due to their microscale diameter, omni−directional flexibility, and scalability into two− or three−dimensional structures [16,17,18]. J.M. Lee et al. reported a high−capacity Ag−Zn yarn battery by embedding active materials into the inner structure of a carbon nanotube (CNT) yarn [19]. H. Li et al. developed a high−performance and stretchable Zn−ion battery using double−helix yarn electrodes [20]. Q. Zhang et al. developed and demonstrated a high−voltage coaxial−fiber−structured rechargeable Zn−ion battery [14]. Moreover, two−dimensional textile−structured batteries have been reported owing to their potential for upscaling, mass production, and commercialization. A.M. Gaikwad et al. [21] and P. Tan et al. [22] demonstrated stretchable MnO_2_−Zn and hybrid−Zn−ion textile−structured batteries with active material deposition onto conductive fabrics, respectively. Y.H. Lee et al. have reported rechargeable textile batteries using electroless deposition of Ni onto a polyester fabric [6].

However, in most of the previously reported textronic batteries, the battery electrodes and electrolyte are exposed to the external environment without a protective outer layer, which causes some critical problems. In practical and actual usage, textronic batteries must withstand varying environmental conditions, such as temperature, humidity, and body sweat, as well as various types of mechanical deformations generated by human body motions, including bending, twisting, rubbing, and pressing, without performance degradations. The external environmental factors can change the composition of the electrolyte, and the mechanical deformations may cause detachment of the electrolyte, leading to failure or malfunction of the textronic battery. Conversely, skin irritation and allergic dermatitis caused by leakage of the acidic or basic electrolyte have been reported previously [23]. Furthermore, washability is one of the significant obstacles to practical applications, especially in the case of yarn− and textile−based devices [13,24]. During laundering, the combined mechanical and chemical actions may rinse away the electrolyte from the textile, which can prevent functioning of and damage to the battery electrodes. Hence, textronic batteries clearly need a protective sheath that can separate the external environment from the internal components. Some pioneering researchers have reported textronic devices with packaging [13,14,16,25], but the complex fabrication process, thick protective layer, and low productivity still pose challenges for their potential use.

In the present study, we developed a wearable and washable MnO_2_−Zn textile battery that maintains its electrochemical capacity under various external environmental conditions with the aid of a vacuum−sealed packaging. The packaged textile battery was fabricated by vacuuming a polymer envelope containing the textile battery made of a commercial textile, followed by heat sealing with a vacuum packaging machine; the textile battery consisted of yarn−structured MnO_2_ and Zn electrodes. During the vacuum packaging process that lasts about 7.5 s, the air inside the envelope was removed, which resulted in a 43% decrease in thickness (from 2.1 to 1.2 mm); further, the interior and exterior regions of the packaged textile battery were completely separated by the packaging sheath to prevent leakage of the gel electrolyte and intrusion of any external substances. Consequently, the resulting packaged textile battery exhibited 96% and 103% of its initial capacity at a temperature of 60 °C and relative humidity (RH) of 90 RH%, respectively, while the nonpackaged textile battery showed a drastic decrease in capacity even under mild conditions of 20 °C and 20 RH%. Various types of applied mechanical deformations from human motions, such as bending, twisting, rubbing, and pressing, resulted in negligible changes to the capacity. Furthermore, we confirmed that several cycles of mechanical, chemical, and a combination of mechanical/chemical washing hardly affected the electrochemical performance. The proposed vacuum−packaged textile battery is thus expected to accelerate the practical use of yarn− or textile−based wearable batteries as well as open up new possibilities for convenient use regardless of varying external conditions.

## 2. Materials and Methods

### 2.1. Chemicals and Materials

The CNT sheets were drawn from well−aligned multiwalled carbon nanotube (MWNT) forests grown by chemical vapor deposition (CVD), with a height of 750 μm (NTAD 10, PDSI Corporation, Korea). MnO_2_ and Zn particles were purchased and used as the active materials of the textile battery (Sigma−Aldrich, Saint Louis, MO, USA). Polyvinyl alcohol (PVA, average Mw = 130,000), zinc chloride (ZnCl_2_), manganese (II) sulfate monohydrate (MnSO_4_), and lithium chloride (LiCl) were used for the gel electrode (Sigma−Aldrich, Saint Louis, MO, USA) and gel electrolyte. A polypropylene (PP) envelope of 125 μm thickness was used for vacuum packaging the textile battery (Cosbig, Incheon, Republic of Korea).

### 2.2. Preparation of MnO_2_ Yarn Cathode, Zn Yarn Anode, Gel Electrolyte, and Textile Battery

The proposed battery with the MnO_2_ yarn cathode and Zn yarn anode was fabricated by the biscrolling method, as previously reported in [26]. Three layers of 7.5 cm (length) × 1 cm (width) CNT sheets were first stacked on glass. To fabricate the MnO_2_ yarn cathode, about 0.1 g of MnO_2_ particles was well dispersed in 5 mL of ethanol by ultrasonication for 30 min and dropped on the prepared CNT sheets. These CNT sheets with MnO_2_ particles were then connected to a custom−made twisting machine with a motor (K6G3C, GGM) and about 400 turns/m of twisting was inserted. After this 2D sheet to 1D yarn cathode fabricated process, MnO_2_ cathode yarn was condensed by ethanol evaporation. Similarly, to fabricate the Zn yarn anode, about 0.1 g of Zn particles was dispersed in 5 mL of ethanol and ultrasonicated for 30 min, following which it was drop cast on the CNT sheet and dried at room temperature for 10 min; the CNT sheet with the Zn particles was then connected to the custom−made twisting machine and about 400 turns/m of twisting was inserted. After yarn anode fabrication, Zn anode yarn was condensed by ethanol evaporation. The gel electrolyte for the wearable textile battery was prepared by mixing PVA (1 g), ZnCl_2_ (5.45 g), MnSO_4_ (1.25 g), and LiCl (2.54 g) in deionized water (20 mL) and stirring at 80 rpm under 90 °C until the mixed solution became transparent. The textile battery was then fabricated with a commercial textile and the prepared cathode and anode yarns battery electrodes. The yarn−structured MnO_2_ cathode and Zn anode were sewn into the commercial textile upside down repeatedly with about 50 mm spacing between the cathode and anode in parallel. The gel electrolyte was then applied on the two parallel sewn yarn batteries in the textile.

### 2.3. Textile Battery Packaging by Vacuum Sealing

Two PP envelope films of 125 μm thickness each were aligned so as to cover the top and bottom surfaces of the MnO_2_−Zn textile battery. Each side of the two PP films was heated and attached to each other to form a one−side−open PP envelope; the last unsealed side was then closed after the vacuuming step once the air in the pouch was removed in about 7.5 s after placing the textile battery inside to minimize the thickness of the packaged textile.

### 2.4. Characterization

Scanning electron microscope (SEM) images of the battery yarn electrodes were obtained (S−4600, Hitachi, Japan), along with optical images of the textile battery using an optical camera (D750, Nikon, Dokyo, Japan). In addition, the electrochemical performances of the nonpackaged and packaged textile batteries were evaluated using an electrochemical analyzer (Vertex EIS, Ivium, Eindhoven, Netherland). The textile battery was packed using a vacuum sealer (Type 5703, Solis, Mendriso, Switzerland).

### 2.5. Calculation

The mass loading weight percent (wt%) of the yarn electrodes was calculated using the following equation:(1)$$\mathrm{wt}\%=\frac{W_{total}-W_{CNT}}{W_{total}}\times 100=\frac{W_{active}}{W_{total}}\times 100\;\left(\mathrm{wt}\%\right)$$
where the *W_total_* is the total weight of the yarn electrodes, *W_CNT_* is the weight of the bare CNT yarn, and the *W_active_* is the weight of the active materials (MnO_2_ for cathode and Zn for anode, respectively) loaded on bare CNT yarn.

The specific capacity (mAh/g) was calculated from the galvanostatic discharging curve using Equation (1):(2)$$Capacity=\frac{I\times \Delta t\;}{m}$$
where *I* is the discharge current of the textile battery, Δ*t* is the time until the discharge voltage reaches 0.5 V, and *m* is the total weight of packaged textile battery comprising the PP film, commercial textile, and MnO_2_−Zn yarn battery electrodes.

## 3. Results

### 3.1. MnO_2_−Zn Textile Battery Packaged by Vacuum Sealing

The vacuum sealing method and resulting wearable and washable MnO_2_−Zn textile battery are schematically illustrated in Figure 1a. Figure 1b,c show the optical images of the MnO_2_−Zn textile battery and PP thin film sheath for packaging (Figure 1c). The MnO_2_−Zn battery is well−known for being ecofriendly and inexpensive, while affording a high discharge voltage and relatively stable cyclic performance [5,17,27,28,29,30,31,32]. In particular, the use of aqueous (water−based) electrolytes in such systems makes them promising alternatives to the inflammable and explosive Li−based batteries used in wearable energy storage devices that are directly placed in contact with human skin. The detailed yarn electrode fabrication process is discussed later. The PP film is well−known for its chemical and thermal resistances (melting temperature between 135 and 160 °C), light weight, and inexpensiveness [33,34]; it is also one of the most widely used materials for laminating and packaging.

Vacuum sealing is a core fabrication method of the proposed wearable and washable textile battery. A textile battery with the sewn cathode and anode yarns was placed in a PP envelope with one open end (Figure 1d(i)). The next step involved vacuuming the interior of the envelope using a commercial vacuum packaging machine (Figure 1d(ii)). In this process, the internal air was removed over a few seconds, leading to a noticeable decrease in thickness by the difference in the air pressure. As shown in Figure 1e, the textile battery exhibited about 43% decrease in its thickness (from 2.1 to 1.2 mm) after vacuum packaging. The detailed thickness changes versus packaging processes are shown in the inset of Figure 1e. While the thickness of the nonpackaged textile battery drastically increased (from 1.1 to 2.1 mm) owing to the residual air in the envelope, a slight increase was observed after the vacuum process (from 1.1 to 1.2 mm). The packaging was finished with heat sealing, where heat energy is applied to the polymer sheath to achieve strong adhesion. The thermal energy acts only on the open edge of the polymer sheath and hardly affected the electrolyte coated on the electrodes and commercial textile. As a result, the interior and exterior of the envelope were completely separated to prevent leakage of the internal substance and intrusion of any external substances (Figure 1f,g).

### 3.2. Electrochemical Performance of the Yarn and Textile MnO_2_−Zn Battery

The MnO_2_ yarn cathode and Zn yarn anode were fabricated by the CNT yarn biscrolling method. The biscrolled yarn electrodes are known to be suitable electrodes for wearable devices and textronics due to their high active−material loading weight and mechanical flexibility. The CNT aerogel sheet has outstanding mechanical and electrical properties while providing a high specific surface area for use as a host material and current collector. The base substrate was prepared by stacking three layers of MWNT sheets of length 7.5 cm and width 1 cm that were drawn from the MWNT forest. The MWNT forest was synthesized by the CVD method, as reported previously [35]. For the MnO_2_ yarn cathode, the MnO_2_ nanoparticles dispersed in ethanol by ultrasonication were drop−cast on the prepared CNT sheet stack, and the MnO_2_ solution−cast CNT sheets were progressively transformed into a yarn−shaped electrode by inserting twists at the rate of 400 turns per meter of the initial sheet length. Lastly, the yarn electrode underwent diameter condensations by surface tension densification during ethanol evaporation, thus effectively embedding the MnO_2_ nanoparticles (i.e., the cathodic active material) inside the yarn. The Zn yarn anode was also prepared using same fabrication process. The diameters of the MnO_2_ yarn cathode and Zn yarn anode after ethanol densification were about 197 and 183 μm, respectively. The mass loading wt% of MnO_2_ yarn cathode and Zn yarn anode were calculated as 69.2 and 60.0 wt%, respectively.

The surface morphologies of the MnO_2_ cathode and Zn anode were investigated next using SEM images, as shown in Figure 2a,b. The images of the yarn electrodes show not only the bias angles between the CNT bundle orientation and yarn longitudinal directions, which were formed by twist insertion during yarn fabrication, but also the uniaxially well−aligned and densely packed CNT bundles with evenly distributed active material nanoparticles on the surface and nano bundle gaps. As shown in the magnified SEM images, such nanoparticle−embedded CNT bundles provide intimate contact between the active materials and current collectors, resulting in an interconnected network structure with highly effective electron pathways during redox reactions [19]. Moreover, the yarn electrodes have outstanding electrical conductivities. The electrical resistances of the MnO_2_ cathode and Zn anode were about 63.9 and 64.8 Ω/cm, respectively (Appendix A). The most remarkable advantage of the biscrolled yarn electrode is its mechanical flexibility, which allows various types of deformations. The biscrolled yarn electrode was bent (Figure 2c(i)) and wound around a glass tube of diameter 1 mm (Figure 2c(ii)) to evaluate flexibility. The yarn electrode was even rolled into the shape of a circle and knotted in the middle by hand, as shown in Figure 2d(i,ii). When applying bending at angles from 30 to 180°, negligible changes were observed in the yarn resistances (Appendix A). These extraordinary mechanical and electrical properties enable hand−woven textile battery systems. Repeated cyclic bending deformations hardly affect the electrochemical performances (Appendix A). About 101.1% and 99.8% of the initial capacity were maintained even after 100 cycles of bending at a bending angle of 180°. As shown in the optical image of the MnO_2_−Zn textile battery configuration (Figure 1b), a full−cell MnO_2_−Zn yarn battery was assembled with the biscrolled MnO_2_ yarn cathode and Zn yarn anode woven in parallel into the commercial textile and coated with 5 wt% of the PVA−ZnCl_2_−MnSO_4_−LiCl gel electrolyte. Since the pressure difference during the vacuum process can cause the leakage, viscous quasi−solid gel electrolyte was exploited instead of liquid electrolyte. The gel electrolyte is required to provide mechanical/electrochemical stability and exchange of reactants under various environmental conditions, including mechanical deformations and temperature and humidity changes, to maintain the electrochemical performance of the battery [19]. The PVA hydrogel, a cross−linked hydrophilic−polymer−based quasi−solid gel electrolyte, is widely used owing to its outstanding ionic conductivity and mechanical flexibility [10,19]. The electrochemical redox reactions for the nonpackaged and packaged MnO_2_−Zn textile batteries were characterized using cyclic voltammetry (CV) and galvanostatic discharge curves with a two−electrode system in which the MnO_2_ and Zn yarn electrodes were used as the cathode and anode, respectively. According to H. Pan [31], the cathodic reaction of the MnO_2_−Zn battery system is given by
$$H_{2}O↔H^{+}+{\mathrm{OH}}^{-}$$
$${\mathrm{MnO}}_{2}+H^{+}+e^{-}↔\mathrm{MnOOH}$$
12Zn2++OH−+16ZnSO4+χ6H2O↔16ZnSO4[Zn(OH)2]3·χH2O
and the anodic reaction is given by
12Zn↔12Zn2++e−

The full−cell redox reaction is thus given by
MnO2+12Zn+χ6H2O+16 ZnSO4↔MnOOH+16ZnSO4[Zn(OH)2]3·χH2O

As shown in Figure 2e, the CV curves of the nonpackaged and packaged MnO_2_−Zn textile batteries measured at a scan rate of 10 mV/s exhibited clear oxidation and reduction peaks centered at 1.68 V and 1.35 V, respectively, showing good agreement with previously reported MnO_2_−Zn battery systems [36,37]. The redox peaks of both textile batteries appeared at the same potential regardless of the vacuum packaging. The galvanostatic discharge curves of the nonpackaged and packaged MnO_2_−Zn textile batteries were measured at a current density of 0.3 mA/g and compared in Figure 2f. Unless otherwise noted, the galvanostatic discharge curves were measured at a discharge current density of 0.3 mA/g. After vacuum packaging, the textile batteries showed negligible differences for both the discharge voltage and capacity. As with the results of the CV curves, both batteries exhibited distinct discharge voltage plateaus between 1.4 and 1.8 V as well as linear capacities (0.238 mAh/g for the packaged and 0.234 mAh/g for the nonpackaged textile batteries). The capacity retention performance of the vacuum−packaged MnO_2_−Zn battery was characterized further for various discharge currents and repeated charge/discharge cycles. Specifically, about 89.8% of the initial capacity was maintained at a discharge current of 1.5 mA/g (inset of Figure 2f), and about 83.3% of the initial capacity was retained after 100 repeated charge/discharge cycles (Appendix A).

### 3.3. Electrochemical Performances under Various Environmental Conditions

Figure 3a is a schematic illustration showing the external environmental factors that affect the electrochemical performance of the wearable and washable MnO_2_−Zn textile battery during practical use. Unexpected environmental conditions such as high temperatures or low humidity and a mixture of water, ions, and various amino acids (also called sweat) can damage the battery electrodes or change the composition of the gel electrolyte, resulting in malfunctions. Conversely, skin inflammation may be caused by leakage of acid− or alkali−based electrolytes, which pose safety issues. Based on the successful development of the wearable and washable MnO_2_−Zn textile battery packaged by vacuum sealing, the dependence of the electrochemical capacity on the temperature and humidity is observed in an oven and a hand−made humidity chamber (Figure 3b). As shown in Figure 3c, the capacity retention performances of the nonpackaged and packaged MnO_2_−Zn textile battery at a temperature of 20 °C and relative humidity of 20 RH% were measured for repeated charge/discharge cycles over 15 h. The packaged textile battery exhibited approximately 97.3% retention of its initial capacity after 15 h. However, the capacity of the unpackaged textile battery decreased drastically after 5 h, showing only 5.9% capacity retention. The advantage of vacuum sealing is thus revealed in the inset of Figure 3c. The gel electrolyte of the nonpackaged battery evaporated and dried out under environmental conditions, but that of the packaged battery remained functional within the polymer sheath. Therefore, it is advantageous to operate the packaged textile battery to obtain stable energy storage performances under various environmental conditions. The galvanostatic curves at various temperatures and humidity conditions are compared in Figure 3d (black line: 20 °C and 20 RH%, red line: 60 °C and 20 RH%, blue line: 20 °C and 90 RH%, green line: 60 °C and 90 RH%). No obvious changes in the capacities were observed for various temperatures and humidity conditions. The dependences of the specific capacities on the temperature and relative humidity are shown in Figure 3e. The electrochemical performances were retained by about 96% and 103% when the temperature increased from 25 to 60 °C and relative humidity increased from 20 to 90 RH%, respectively.

### 3.4. Electrochemical Performances under Various Mechanical Deformations

During practical and actual usage, textile batteries are exposed to various types of mechanical deformations caused by human movements (Figure 4a). In numerous human skeletal muscles and joints, bending and twisting deformations are generated such that friction is commonly applied at the human skin, such as in the brachial, femoral, and lateral abdominal regions. Furthermore, the plantar region bears tremendous pressures beyond the bodyweight during walking or running; these deformations can cause major performance degradations to the textile battery. Thus, stable electrochemical performance during movement is another essential requirement of a textile battery. The reversible deformability and flexible polymer sheath allow our packaged textile battery to maintain its retention performance, showing great promise for practical applications. The galvanostatic discharge curves for the pristine (black line), bent at an angle of 180° (red line), twisted at an angle of 180° (blue line), rubbed at a frequency of 2 Hz (green line), and pressed with 7.5 kPa (orange line) packaged textile batteries are compared in Figure 4b. No significant differences were observed in the specific capacities and discharge voltages. The capacities were measured, and retention performances calculated under different bending and twisting angles from 0 to 180° to demonstrate the stability of the device under harsh bending and twisting (Figure 4c). At the maximum bending and twisting angles (180°), the specific capacities retained were about 87% and 91% of the initial capacity, respectively; impressive capacity retentions were thus observed for both bending and twisting deformations. Moreover, the packaged textile battery was periodically rubbed at a frequency of 2 Hz (Figure 4d) and compressed with gradually increasing pressures from 2.5 to 7.5 kPa before recovering to the pristine state (Figure 4e) during the galvanostatic evaluations. During the dynamically applied rubbing and pressing deformations, the textile battery continued to exhibit stable discharge voltages.

### 3.5. Electrochemical Performances during Washing

Considering the practical applications for textronics, washability is one of the most important and distinct characteristics. Laundering involves the mechanically and chemically complex actions of washing and rinsing to remove contaminants from textile products. During laundering, detergents containing surfactants chemically dissolve the greasy dirt in water and the applied external force mechanically removes contaminants from the textile. We simulated the laundering process of contaminated textile battery by mechanical washing with a rotating motion and chemical washing with a detergent solution (Figure 5a,b). As shown in Figure 5c, the specific capacity showed negligible changes during five cycles of mechanical washing at 100 rpm and five cycles of chemical washing by immersion of the textile battery in a detergent solution for 10 min each time. Then, the mechanical and chemical washing actions were applied simultaneously for five cycles; these multiple washing processes demonstrate no noticeable changes in the electrochemical performance of the packaged textile battery.

## 4. Conclusions

In summary, a wearable and washable MnO_2_−Zn textile battery packaged by the vacuum sealing method was demonstrated in this study. The textile battery was composed of a biscrolled MnO_2_ yarn cathode and a Zn yarn anode, a commercial textile substrate, and an aqueous gel electrolyte. After packaging by vacuum sealing, the interior and exterior regions of the packaged textile battery were completely separated to preclude leakage and intrusion of substances within the packaging sheath. The resulting packaged textile battery exhibited 96% and 103% of its initial capacity at a high temperature of 60 °C and relative humidity of 90 RH%, respectively, while the nonpackaged textile battery underwent drastic capacity degradation even under mild conditions of 20 °C and 20 RH%. Stable capacity retention performances were also observed regardless of mechanical deformations, including bending, twisting, rubbing, and pressing. Furthermore, the electrochemical capacity was well maintained after several cycles of mechanical washing at 100 rpm, chemical washing by immersing the textile battery in a detergent solution for 10 min each time, and a combination of mechanical and chemical washing. On the basis of these demonstrations, vacuum packaging of the textile battery is shown to overcome the limitations of existing works, in which the battery electrodes and electrolyte are exposed to the external environment. We expect that our vacuum−packaged textile battery can accelerate the practical use of wearable batteries and open up new possibilities for convenient use of textronics.

## Figures and Tables

**Figure 1 nanomaterials-13-00265-f001:**
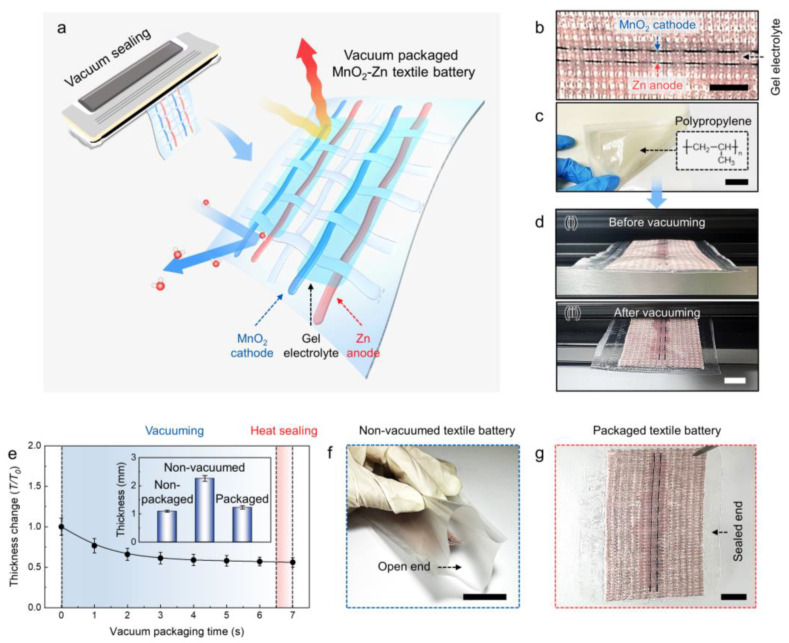
(**a**) Schematic illustration of the vacuum−packaged MnO_2_−Zn textile battery (blue yarns for MnO_2_ cathode and red yarns for Zn anode) assembled with a vacuum packaging machine. Optical microscope images showing (**b**) nonpackaged MnO_2_−Zn textile battery and (**c**) polypropylene film used to package the MnO_2_−Zn textile battery (scale bars are 1 and 2 cm, respectively). (**d**) Actual images of the textile battery (**i**) before and (**ii**) after vacuum packaging (scale bar = 2 cm). (**e**) Measured thickness changes (*T*/*T*_0_) versus vacuum packaging time until heat sealing (*T*_0_ represents the thickness of the nonvacuumed textile battery); the inset graph shows the thicknesses of the nonpackaged, nonvacuumed, and packaged textile batteries. Optical photographs of the (**f**) nonpackaged (before vacuuming and heat sealing) and (**g**) packaged (after vacuuming and heat sealing) battery (scale bars are 3 and 1 cm, respectively).

**Figure 2 nanomaterials-13-00265-f002:**
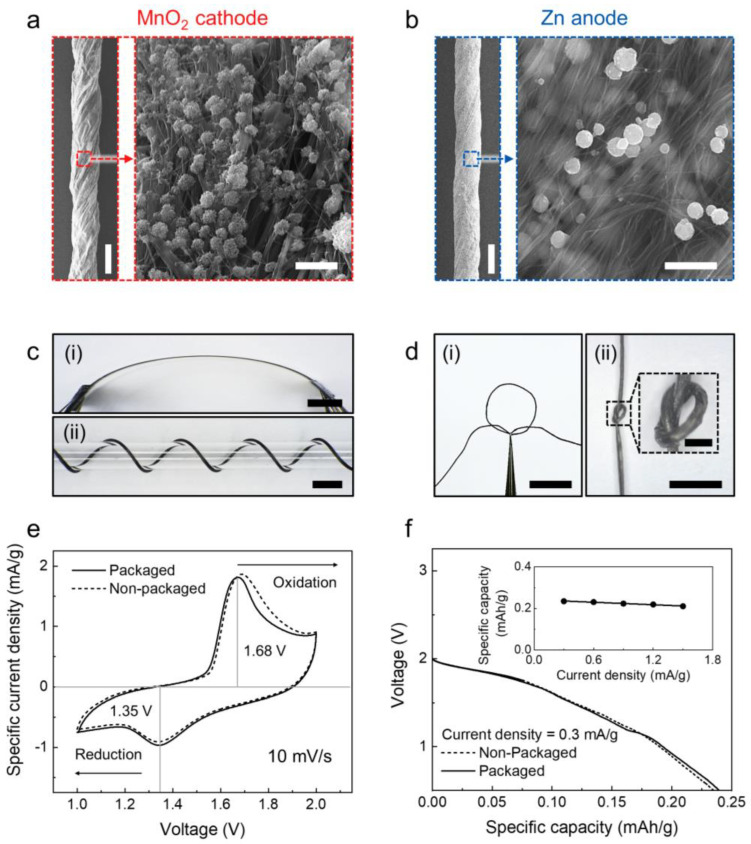
Scanning electron microscope (SEM) images of the (**a**) MnO_2_ cathode (scale bars are 200 and 0.5 μm, respectively) and (**b**) Zn anode (scale bars are 200 and 0.5 μm, respectively) showing the entire structures and yarn surfaces of the battery electrodes. Optical images showing various mechanical deformations such as (**c**) (**i**) bending (scale bar = 5 cm) and (**ii**) winding (scale bar = 2 cm), and (**d**) (**i**) circling (scale bar = 5 cm) and (**ii**) knotting (scale bars are 2 cm and 400 μm, respectively). (**e**) Cyclic voltammetry (CV) curves of the packaged and nonpackaged MnO_2_−Zn batteries in a quasi−solid−state aqueous electrolyte measured at a scan rate of 10 mV/s. (**f**) Galvanostatic discharge curves of the nonpackaged and packaged textile batteries; the inset graph shows the dependence of the specific capacity on current density from 0.3 to 1.5 mA/g.

**Figure 3 nanomaterials-13-00265-f003:**
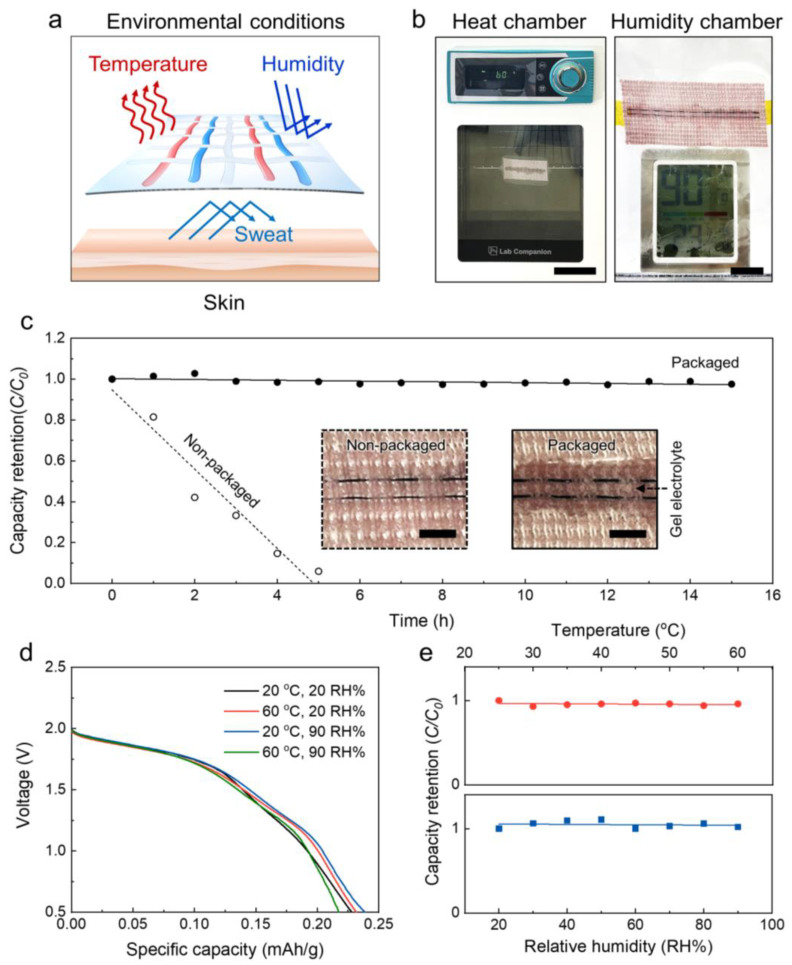
(**a**) Schematic illustration showing the external environmental factors that affect the electrochemical performance of the textile battery. Optical photographs of the (**b**) oven and (**c**) humidity chambers in the experiments (scale bars are 8 and 2 cm, respectively). (**c**) Capacity retention performances of the nonpackaged and packaged textile batteries for 15 h; the inset optical images show the nonpackaged and packaged textile batteries exposed to environmental conditions for 5 h. (**d**) Galvanostatic discharge curves of the packaged textile battery at 20 °C and 20 RH% (black line), 60 °C and 20 RH% (red line), 20 °C and 90 RH% (blue line), and 60 °C and 90 RH% (green line). (**e**) Capacity retention curves versus temperature (**top**) and relative humidity (**bottom**).

**Figure 4 nanomaterials-13-00265-f004:**
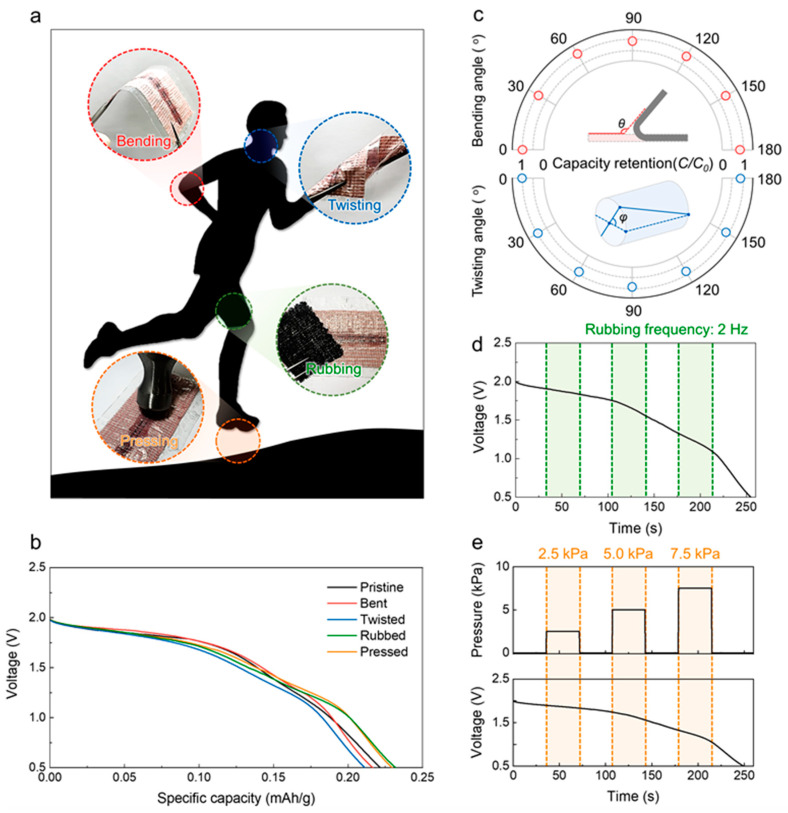
(**a**) Schematic illustrations and optical images showing human motions and evoked mechanical deformations, including bending, twisting, rubbing, and pressing. (**b**) Galvanostatic discharge curves of the textile battery under pristine, bent, twisted, rubbed, and pressed states. (**c**) Capacity retention versus bending and twisting angles; the inset illustrations exhibit the definitions of the bending (*θ*) and twisting (*φ*) angles. Galvanostatic discharge curves under dynamically applied (**d**) rubbing with a frequency of 2 Hz and (**e**) pressing with deformation pressures of 2.5, 5.0, and 7.5 kPa.

**Figure 5 nanomaterials-13-00265-f005:**
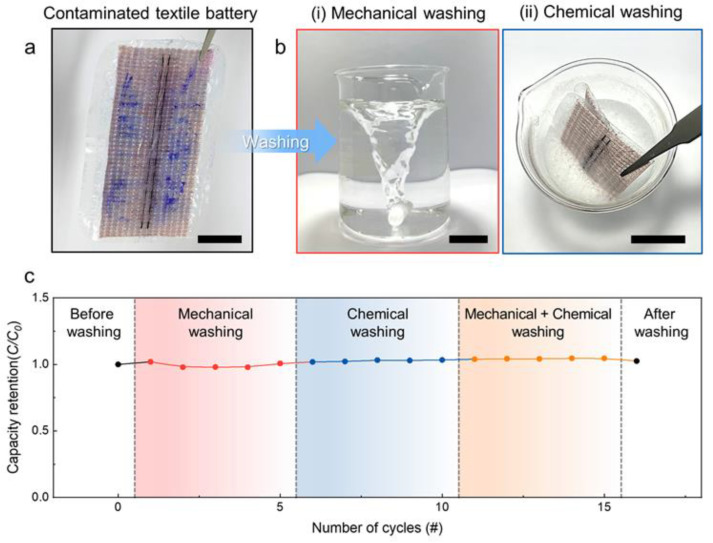
Optical photographs showing the (**a**) contaminated vacuum−packaged MnO_2_−Zn textile battery and (**b**) steps for (**i**) mechanical and (**ii**) chemical washing (all scale bars are 2 cm). (**c**) Capacity retention for the three distinct types of washing: mechanical, chemical, and combined mechanical and chemical.

## Data Availability

Not applicable.

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
