# Peer review of "Wearable and Washable MnO2−Zn Battery Packaged by Vacuum Sealing"

_nanomaterials, 2023, doi:10.3390/nano13020265_

Round 1

Reviewer 1 Report

This manuscript demonstrated a wearable and washable MnO2-Zn textile battery that maintains its electrochemical capacity under various external environmental conditions through vacuum-sealed packaging. The resulting packaged textile battery exhibits stable capacity retention performance under varying temperatures and humidity. The performance is good. The paper is well written. I recommend a minor revision. Detailed comments can be seen below:

1.       It would be clearer if the author could demonstrate more about adding electrolytes to the system. For example, are the electrodes warped by gel electrolytes before sealing? How to prevent the electrolyte from vacuum drying when packing?

2.       Please point out where the electrolyte is in Figure 1a.

3.       The arrow direction of the oxidation and reduction in Figure 2e is not right. Please further check.

4.       What are the mass loadings of the MnO2 cathode and Zn anode?

5.       Some high-impact papers regarding flexible Zn batteries or fiber-shaped devices are not cited. For example, Advanced Materials 31 (36), 1903675; Advanced Materials 29 (26), 1700274; Advanced Materials 29 (44), 1702698; Small Methods 3 (12), 1900525; Journal of Materials Chemistry A 6 (26), 12250-12258.

Author Response

We appreciate the positive comments of the reviewers and the suggestions of the editor. The manuscript and supplemental materials have been accordingly revised to clearly address the issues raised by reviewers. To help address the comments of the reviewers, we added pertinent explanations and results. This includes (1) additional description of gel electrolyte coating, (2) revised CV curves, and (3) calculation of active material loading weight percent. According to the comments from the reviewer, the recent papers were newly added in reference section.

Reviewer 2 Report

This manuscript deals with a wearable MnO2-Zn battery packaged by vacuum sealing for practical use of textronics. In he introduction the authors justify their interest in this study and include 22 references that help potential readers to understand the basics of textile batteries and their practical importance, as well as the steps adopted for the authors to attain their objectives. In section 2. on materials and methods, the required chemicals and materials are reported, then the preparation of the MnO2 yarn cathode, Zn yarn anode, gel electrolyte, and textile battery, are described in a brief but clear and proper way. The section also includes the  textile battery packaging by vacuum sealing, the techniques used to characterize the system (SEM, Electrochemicl analyzer) and the GDC used to calculate the electrochemical specific capacity. The third section deals with the results and their analysis an discussion. The first figure illustrates the vacuum packaged battery that helps understanding the involved steps performed to prepare the sealed battery. Figure 2 hows the SEM images of cathode, anode, entire structures and yarn surfaces of the electrodes; the figure also includes CVs and GD curves. Figure 3 illustrates the external environmental factors that affect the electrochemical behaviour of the battery. The analysis continues by considering the electrochemical performance of the yarn and textile battery. The approach provides clear and deep information confirming the ability of the authors in dealing with this subject of highly technological importance. Electrochemical performances under various environmental conditions, mechanical deformations, and during washing, are also discussed being complemented by figures 4 and 5, that illustrate human motions and evoked mechanical deformations, as well as the laundering process of contaminated textile battery by mechanical and chemical washing. In summary, an excellent work deserving publication. 

Author Response

We appreciate the positive comments of the reviewers and the suggestions of the editor.

Reviewer #2

Comments and Suggestions for Authors

This manuscript deals with a wearable MnO2-Zn battery packaged by vacuum sealing for practical use of textronics. In the introduction the authors justify their interest in this study and include 22 references that help potential readers to understand the basics of textile batteries and their practical importance, as well as the steps adopted for the authors to attain their objectives. In section 2. on materials and methods, the required chemicals and materials are reported, then the preparation of the MnO2 yarn cathode, Zn yarn anode, gel electrolyte, and textile battery, are described in a brief but clear and proper way. The section also includes the textile battery packaging by vacuum sealing, the techniques used to characterize the system (SEM, Electrochemical analyzer) and the GDC used to calculate the electrochemical specific capacity. The third section deals with the results and their analysis and discussion. The first figure illustrates the vacuum packaged battery that helps understanding the involved steps performed to prepare the sealed battery. Figure 2 shows the SEM images of cathode, anode, entire structures and yarn surfaces of the electrodes; the figure also includes CVs and GD curves. Figure 3 illustrates the external environmental factors that affect the electrochemical behaviour of the battery. The analysis continues by considering the electrochemical performance of the yarn and textile battery. The approach provides clear and deep information confirming the ability of the authors in dealing with this subject of highly technological importance. Electrochemical performances under various environmental conditions, mechanical deformations, and during washing, are also discussed being complemented by figures 4 and 5, that illustrate human motions and evoked mechanical deformations, as well as the laundering process of contaminated textile battery by mechanical and chemical washing. In summary, an excellent work deserving publication. 

Response: We really appreciate for your positive comment and interest in our work.
